# Multiplexed genetic engineering of human hematopoietic stem and progenitor cells using CRISPR/Cas9 and AAV6

Rasmus O Bak[1†‡§], Daniel P Dever[1†], Andreas Reinisch[2,3,4†], David Cruz Hernandez[2,3,4], Ravindra Majeti[2,3,4]*, Matthew H Porteus[1]*

[1]Department of Pediatrics, Stanford University, Stanford, United States; [2]Department of Medicine, Division of Hematology, Stanford University, Stanford, United States; [3]Department of Medicine, Institute for Stem Cell Biology and Regenerative Medicine, Stanford University, Stanford, United States; [4]Department of Medicine, Cancer Institute, Stanford University, Stanford, United States

*For correspondence: rmajeti@stanford.edu (RM); mporteus@stanford.edu (MHP)

[†]These authors contributed equally to this work

Present address: [‡]Department of Biomedicine, Aarhus University, Aarhus, Denmark; [§]Aarhus Institute for Advanced Studies (AIAS), Aarhus University, Aarhus, Denmark

**Abstract** Precise and efficient manipulation of genes is crucial for understanding the molecular mechanisms that govern human hematopoiesis and for developing novel therapies for diseases of the blood and immune system. Current methods do not enable precise engineering of complex genotypes that can be easily tracked in a mixed population of cells. We describe a method to multiplex homologous recombination (HR) in human hematopoietic stem and progenitor cells and primary human T cells by combining rAAV6 donor delivery and the CRISPR/Cas9 system delivered as ribonucleoproteins (RNPs). In addition, the use of reporter genes allows FACS-purification and tracking of cells that have had multiple alleles or loci modified by HR. We believe this method will enable broad applications not only to the study of human hematopoietic gene function and networks, but also to perform sophisticated synthetic biology to develop innovative engineered stem cell-based therapeutics.

DOI: https://doi.org/10.7554/eLife.27873.001

## Introduction

The current gold standard method for studying human hematopoietic stem and progenitor cell (HSPC) gene function has been either overexpression or RNAi-mediated knockdown of genes using lentiviral vectors (*Doulatov et al., 2012*; *Chan et al., 2015*). While these methods have provided great insights into HSPC biology, they come with several confounders, such as random integration of the vector into the host genome, unregulated transgene expression, and incomplete gene knockdown (*Woods et al., 2006*; *Naldini, 2015*). More recently, programmable nucleases such as zinc finger nucleases (ZFNs), transcription activator-like effector nucleases (TALENs), and CRISPR/Cas9 have been utilized to disrupt genes by the introduction of site-specific DNA double strand breaks (DSBs) that are corrected through non-homologous end-joining (NHEJ) (*Hendel et al., 2015*; *Holt et al., 2010*; *Saydaminova et al., 2015*; *Mandal et al., 2014*; *Schumann et al., 2015*; *Kim et al., 2014*; *Lin et al., 2014*). This error-prone system creates a heterogeneous mixture of cells with various genotypes of SNPs and small insertions or deletions (INDELs); moreover, not all of the genetic changes from INDELs cause functional gene disruption as they may preserve the open reading frame and may not change amino acids essential for protein functions (*Shi et al., 2015*; *Hultquist et al., 2016*). In a prior study, defined gene deletions were created in HSPCs using a dual sgRNA approach, however, more than half of the alleles were not modified leading to residual gene expression

**eLife digest** Our DNA contains thousands of sections called genes that encode the information needed to make all the cells in the human body. To understand what the genes do and how they contribute to diseases, it is crucial for researchers to be able to switch individual genes on or off or make precise changes to the 'letters' in their code. Since most genes act in complicated networks it would be very useful to be able to edit several genes at the same time, especially when studying cancer and other diseases that are caused by defects in multiple genes.

CRISPR/Cas9 is a relatively new technique that allows the code of individual genes to be precisely edited. To edit a gene, CRISPR/Cas9 first breaks the DNA at the site of interest and this break is subsequently repaired using new DNA templates that introduce the desired change in the code. In this way, the letters of the code can be changed with the same precision that one edits the letters and words of a document. This technique has been successfully used to edit the code of single genes, but it is much more difficult to use it to edit several genes at the same time.

To import new DNA repair templates into human and other mammalian cells, researchers have used harmless virus-like particles called rAAV vectors. Researchers load the DNA templates into rAAV vectors, which are able to enter the cells and carry the templates to the DNA of the cells. Bak, Dever, Reinisch et al. combined CRISPR/Cas9 with rAAV template delivery to precisely edit several genes in human cells, including blood stem cells. In this new system, CRISPR/Cas9 directs the insertion of new pieces of DNA carried by rAAV6 vectors into specific genes.

The system developed by Bak, Dever, Reinisch et al. allows several genes to be precisely edited at the same time. Furthermore, the system includes fluorescent markers that enable successfully edited cells to be identified and tracked. In the future, this technique could be used to study how genes work together to control various characteristics, and how cancer and other diseases develop.
DOI: https://doi.org/10.7554/eLife.27873.002

(*Mandal et al., 2014*). Another limitation of this prior study is that successfully modified cells were not distinguishable from unmodified wild type (WT) cells, and therefore could not be tracked or isolated as an enriched population. Although the versatility of the CRISPR/Cas9 system allows for simultaneous manipulation at multiple genetic loci in a single cell, multiplexing of NHEJ-based gene editing has mainly been performed in immortalized human cancer cell lines and mouse cells (*Hultquist et al., 2016*; *Cong et al., 2013*; *Heckl et al., 2014*; *Platt et al., 2014*; *Brown et al., 2016*). Finally, these interesting multiplexed proof-of-concept studies, only used NHEJ-mediated editing and did not harness the power of homologous recombination (HR) to create more sophisticated alterations to the genome at multiple alleles and/or loci.

Here, we report an HR-mediated genome engineering method in human HSPCs and T cells that overcomes these limitations and enables the generation and enrichment of HSPC or T cell populations with complete gene knockout or gene replacement at multiple genetic loci. This method has the power to reveal functional gene networks during hematopoiesis and immune system disease pathogenesis and could be combined with the concepts of synthetic biology to create novel stem cell based therapeutics.

## Results

### Enriching HSPCs with targeted integration

We and others have previously shown that HR in human HSPCs can be efficiently induced by site-specific nucleases in combination with homologous donor DNA delivered as single-stranded oligonucleotides (ssODNs), integration-defective lentiviral vectors (IDLVs), or by recombinant adeno-associated virus serotype 6 (rAAV6) vectors (*Dever et al., 2016*; *DeWitt et al., 2016*; *De Ravin et al., 2017*; *Wang et al., 2015*; *Hoban et al., 2016*). We previously showed targeted integration in the beta-globin gene (*HBB*) by combining delivery of Cas9 protein pre-complexed with chemically modified sgRNAs (RNP) and delivery of an AAV6 donor. After successful on-target integration of a reporter transgene, FACS-based sorting of transgene reporter^high-expressing HSPCs was used to purify an HSPC population with >90% targeted integration that displayed long-term repopulation

capacity in NSG mice (*Dever et al., 2016*). To extend this method beyond the *HBB* locus for therapeutic genome editing approaches of hemoglobinopathies, we tested six additional loci for their potential to be modified through HR by CRISPR/Cas9 in combination with AAV6-derived donor delivery. These genes are associated with hematopoiesis, hematopoietic malignancies, or safe harbor sites and include: interleukin-2 receptor gamma chain (*IL2RG*), chemokine (C-C motif) receptor 5 (*CCR5*), runt-related transcription factor one isoform c (*RUNX1c*), additional sex combs like 1 (*ASXL1*), stromal antigen 2 (*STAG2*), and adeno-associated virus integration site 1 (*AAVS1*) (*Tebas et al., 2014*; *Genovese et al., 2014*; *Patel et al., 2012*; *Mazumdar et al., 2015*; *Kotin et al., 1992*). Following electroporation with Cas9 RNP, containing a chemically-modified sgRNA targeting a single site in the selected locus, and transduction with an rAAV6 donor vector carrying homology arms for the targeted site and an expression cassette encoding a fluorescent reporter gene (*Figure 1—figure supplement 1a*), we observed at early time points (day 4) a cell population with increased fluorescence intensity detectable by flow cytometry (reporter$^{high}$ cells) compared to cells receiving only the rAAV6 donor without electroporation of Cas9 RNP (reporter$^{low}$) (*Figure 1a* and *Supplementary file 1a*). For cells targeted at either *CCR5* or *IL2RG*, reporter$^{high}$, reporter$^{low}$, and reporter$^{neg}$ populations were sorted at day four post-electroporation and cultured up to 22 days. Reporter$^{high}$ populations remained 99.2 ± 0.7% reporter positive (*Figure 1b*) while sorted reporter$^{low}$ and reporter$^{neg}$ populations were 29.3 ± 5.4% and 0.6 ± 0.2% reporter positive, respectively. Dividing the reporter$^{low}$ cells into three sub fractions based on fluorescence intensity revealed that GFP intensity at day four post-electroporation positively correlated with the propensity for maintaining GFP expression at day 20 (*Figure 1—figure supplement 1b–c*). In addition, single reporter$^{high}$ cells were plated in methylcellulose to assess integration events at the clonal level. Targeted HSPCs formed a mix of myeloid (CFU-M/GM) and erythroid colonies (BFU-E, CFU-E) indicating that they retained HSPC function. 'In-Out PCR' (one donor-specific primer and one locus-specific primer outside of the respective homology arms) on genomic DNA (gDNA) from single cell-derived methylcellulose colonies confirmed that 99%, 92%, and 100% of reporter$^{high}$ HSPCs targeted at *CCR5* (338 clones analyzed), *IL2RG* (117 clones analyzed), and *RUNX1* (36 clones analyzed), respectively, had at least a monoallelic targeted integration (*Figure 1c* and *Figure 1—figure supplement 2*). Analyses of clones with only mono-allelic integration showed gene-specific differences in the modification of the non-integrated alleles ranging from 38% INDELs for *IL2RG* to 89% INDELs for *CCR5*% and 88% INDELs for *RUNX1*, among which the majority was gene-disrupting (*Figure 1—figure supplement 2* and *Supplementary file 1b*). Collectively, these data indicate that the observed log-fold transgene expression shift following rAAV6 and RNP delivery is due to HR at the intended locus and that reporter expression can be used to enrich gene-targeted HSPCs.

To evaluate the applicability of this technology in a biologically relevant setting we decided to modify the cohesin complex member, *STAG2,* in primary CD34$^+$ HSPCs. The cohesin complex has previously been shown to play an essential part in maintaining normal erythroid differentiation potential of hematopoietic stem and progenitor cells (*Mazumdar et al., 2015*; *Viny et al., 2015*; *Mullenders et al., 2015*). Since the *STAG2* gene is located on the human X chromosome, single-allele integration of a fluorescent reporter in male cells would be sufficient to fully knock out the gene. As expected, Cas9 RNP combined with rAAV6 donor transduction resulted in the generation of a reporter$^{high}$ population that could be sorted for subsequent differentiation experiments. Single cell methylcellulose assays of reporter$^{high}$ cells revealed an almost complete loss in the capacity to form erythroid colonies compared to cells that had only been exposed to rAAV6 and not Cas9 RNP, and also compared to cells with targeted integration at the *AAVS1* locus (*Figure 1d*). These proof-of-concept studies provide evidence that gene-specific enrichment of reporter$^{high}$ cells can be used to study HSPC gene function.

## Biallelic targeted integration in HSPCs

To determine if this method could be used to enrich HSPCs with biallelic gene disruption, necessary for complete functional gene knockout, we targeted the *ASXL1* gene and simultaneously provided GFP and BFP-encoding rAAV6 donors. Four days after electroporation and transduction, 10.4% of cells were double positive for GFP$^{high}$ and BFP$^{high}$ compared to 0.2% for the AAV only sample (*Figure 2a*). Similarly, double-positive populations were apparent when targeting three other genes (*RUNX1*, *HBB*, and *CCR5)* with two rAAV6 donors with various color combinations (*Figure 2—figure supplement 1* and *Supplementary file 1c*). Double-positive cells sorted at day four after

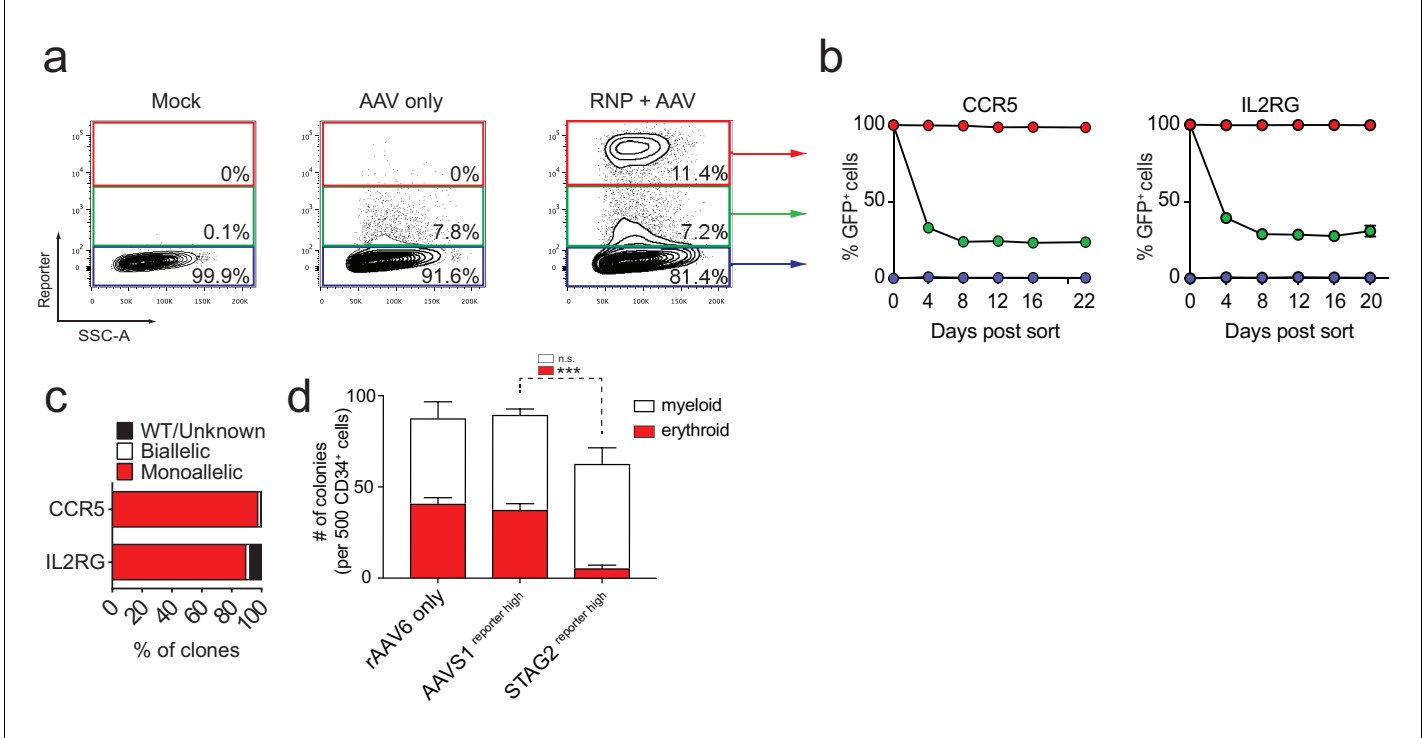

**Figure 1.** FACS-based identification and enrichment of monogenic genome-edited CD34[+] human hematopoietic stem and progenitor cells (HSPCs). (a) HSPCs were electroporated with *CCR5*-RNP and transduced with *CCR5*-tNGFR rAAV6 HR donor. Representative FACS plots from day four post-electroporation highlight the *CCR5* tNGFR[high] population (red gate) generated by the addition of Cas9 RNP compared to cells with low reporter expression (green gate) and reporter[negative] cells (black gate). Numbers reflect percentage of cells within gates. (b) Day four post-electroporation, *CCR5* (tNGFR or GFP) and *IL2RG* (GFP)-targeted HSPCs from reporter[high] (red), reporter[low] (green), and reporter[neg] (blue) fractions were sorted and cultured for 20-22 days while monitoring the percentage of cells that remained GFP[+]. Error bars represent S.E.M. *N* = 6 for *CCR5*, *N* = 3 for *IL2RG*, all from different CD34[+] donors. (c) HSPCs were targeted at *CCR5* (with GFP or tNGFR donor) or at *IL2RG* (GFP donor; only female cells for *IL2RG*). At day four post-electroporation, reporter[high] cells were single-cell sorted into methylcellulose for colony formation. PCR was performed on colony-derived gDNA to detect targeted integrations. 338 *CCR5* and 177 *IL2RG* myeloid and erythroid methylcellulose colonies were screened from at least two different CD34[+] HSPC donors. (d) HSPCs were targeted at the *STAG2* gene or the *AAVS1* locus with a GFP reporter cassette. Cells that only received the *STAG2*-GFP AAV6 donor and not Cas9 RNP were included as an additional control. At day four post-electroporation and transduction, reporter[high] cells from the *STAG2* and *AAVS1* targeting experiments and bulk cells from the *STAG2* AAV6 only population were plated in methylcellulose for colony formation. After 14 days, colonies were scored as either erythroid or myeloid based on morphology. Error bars represent S.E.M, *N* = 3, ***p<0.001, n. s. = p≥0.05, unpaired t-test.

DOI: https://doi.org/10.7554/eLife.27873.003

The following figure supplements are available for figure 1:

**Figure supplement 1.** Analysis of cell fractions with different fluorescence intensity.
DOI: https://doi.org/10.7554/eLife.27873.004
**Figure supplement 2.** Genotypes of clones with mono-genic targeting.
DOI: https://doi.org/10.7554/eLife.27873.005

electroporation remained 94% double-positive for more than two weeks in culture (*Figure 2b*). 'In-out PCR' on gDNA from single cell-derived methylcellulose clones confirmed on-target integration of one transgene into one allele and the other transgene into the second allele (*Figure 2c*). We next tested if the biallelic targeting approach could be extended to another blood cell type and therefore targeted primary human T cells for biallelic HR at *CCR5*. After electroporation with *CCR5*-targeting Cas9 RNP followed by transduction with GFP and mCherry *CCR5* rAAV6 donors, a GFP[high]/mCherry-[high] double-positive population was observed, indicative of biallelic integration at the *CCR5* gene (*Figure 2d*). No significant toxicity was associated with biallelic targeting in T cells (*Figure 2—figure supplement 2*). Overall, these results demonstrate the utility of using rAAV6, Cas9 RNP, and FACS to enrich for primary human HSPCs and T cells that have undergone biallelic homologous recombination, which may have applications for studying hematological and immunological diseases or

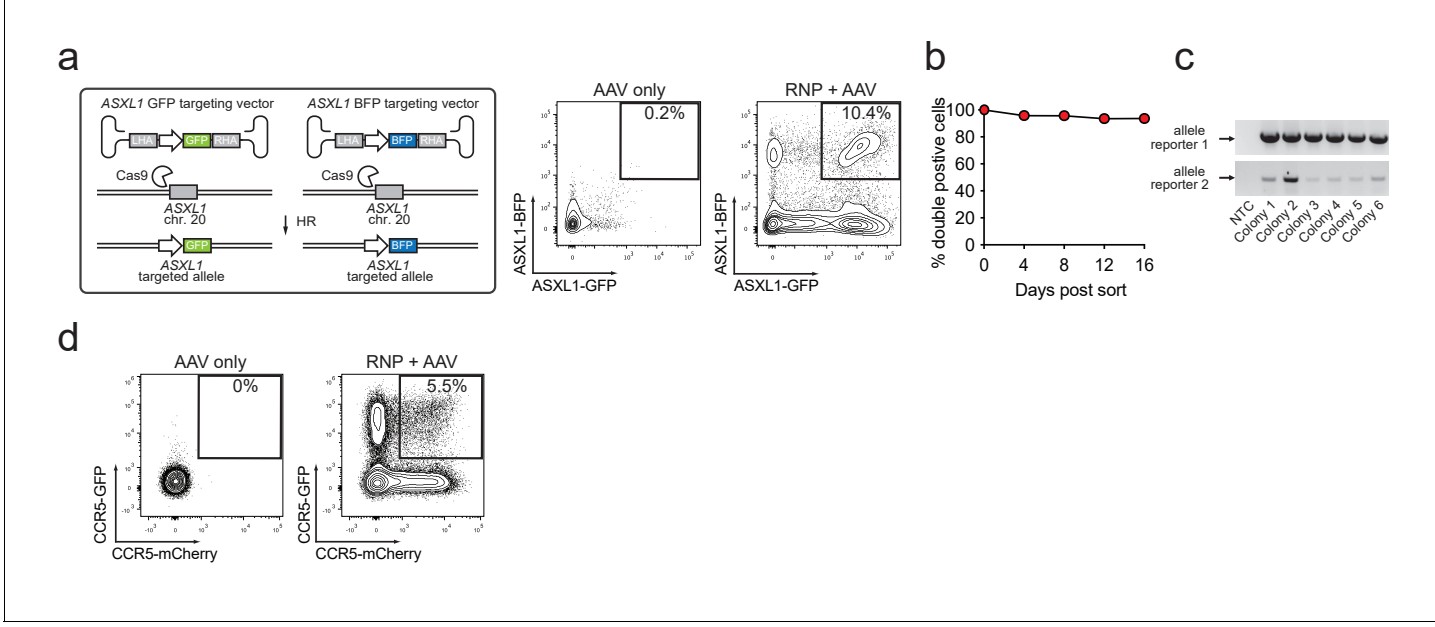

**Figure 2.** Identification and enrichment of biallelic genome-edited CD34+ human hematopoietic stem and progenitor cells (HSPCs). (**a**) *Left,* Schematic showing biallelic targeting strategy for *ASXL1* using GFP and BFP-encoding rAAV6 donors for integration into each allele of *ASXL1*. The SFFV promoter drives reporter expression. *Middle, FACS plot from an* 'AAV only' sample day four post electroporation, showing low episomal reporter expression (BFP and GFP) in cells without the CRISPR system. *Right,* FACS plot of CD34+ HSPCs treated with both Cas9 RNP and the two rAAV6 donors highlighting the generation of BFP[high]/GFP[high] double positive cells that have undergone *ASXL1* dual-allelic targeting. (**b**) HSPCs were targeted at both alleles of *HBB* (Cas9 RNP with GFP and tdTomato rAAV6 donors) and at day four post electroporation, dual positive cells were sorted and cultured for 16 days while analyzing reporter expression. Error bars representing S.E.M. are present, but too small to be visible (*N* = 3 different HSPC donors). (**c**) Gel images showing PCR genotyping of six methylcellulose-derived clones from (**e**) confirming integration into each of the *HBB* alleles. (**d**) Human primary T cells were CD3/CD28 stimulated for three days and then electroporated with *CCR5*-targeting Cas9 RNP and transduced with two *CCR5*-specific rAAV6 donors encoding GFP and mCherry, respectively. FACS plots show GFP[high]/mCherry[high] biallelic targeting frequencies at day four post-electroporation.
DOI: https://doi.org/10.7554/eLife.27873.006

The following figure supplements are available for figure 2:

**Figure supplement 1.** Cas9 and rAAV6-mediated biallelic homologous recombination (HR) in human CD34+HSPCs.
DOI: https://doi.org/10.7554/eLife.27873.007

**Figure supplement 2.** Toxicity assessment of biallelic integration at the *CCR5* locus in primary human T cells.
DOI: https://doi.org/10.7554/eLife.27873.008

generating HSPC or T cell therapeutics that require gene modifications or gene knockout at both alleles.

## Simultaneous HR-mediated targeting of two genes (Di-Genic) in HSPCs

The vast majority of hematopoietic functions and immune diseases are governed by complex, polygenic networks (*Seita and Weissman, 2010*). To potentially study gene-gene interactions and/or generate cell therapeutics with HR modifications at two separate genes, we tested whether our methodology could facilitate simultaneous di-genic (two different genes) HR in HSPCs. We therefore co-delivered *HBB*-tdTomato and *IL2RG*-GFP rAAV6 donors with Cas9 RNP targeting both genes. This strategy produced 10.2% double positive GFP[high]/tdTomato[high] HSPCs compared to 0.1% for the AAV only control sample (*Figure 3a*). We also generated double reporter[high] positive populations when testing other combinations of di-genic HR (*IL2RG/CCR5, RUNX1/ASXL1*, and *HBB/CCR5*) (*Figure 3—figure supplement 1* and *Supplementary file 1c*). Again, double reporter[high] positive cells sorted at day four post-electroporation remained 94% double positive for 15 days in culture (*Figure 3b*). 'In-Out PCR' on double positive methylcellulose myeloid and erythroid clones showed on-target integration at both loci in 88% of clones (57 clones analyzed) (*Figure 3c and d*).

Since the combination of two sgRNAs has previously been used to create and study oncogenic translocations (*Maddalo et al., 2014*), and multiplexed TALEN-mediated gene editing in primary

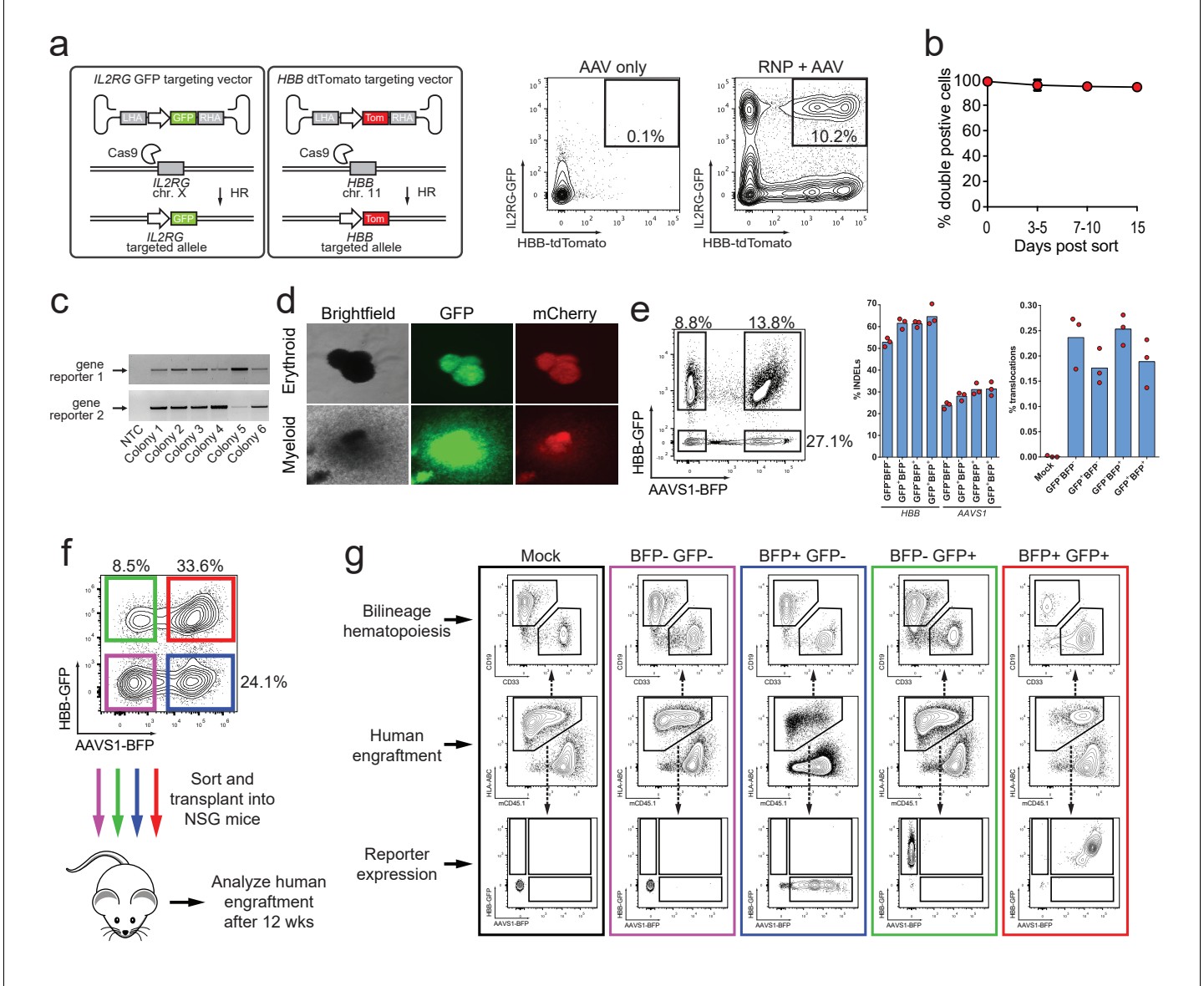

**Figure 3.** Identification, enrichment, and long-term engraftment in NSG mice of di-genic genome-edited CD34+ human hematopoietic stem and progenitor cells (HSPCs). (a) *Left*, Schematic depicting *HBB* and *IL2RG* di-genic targeting. *Middle*, FACS plot of an 'AAV only' sample at day four post electroporation, showing low episomal reporter expression (*HBB*-tdTomato and *IL2RG*-GFP) in cells without the CRISPR system. *Right*, FACS plot at day four post-electroporation of HSPCs electroporated with Cas9 RNP targeting both *HBB* and *IL2RG* followed by transduction with *HBB*-tdTomato and *IL2RG*-GFP rAAV6 donors showing the generation of tdTomato^high^/GFP^high^ cells with di-genic targeting at *HBB* and *IL2RG*. (b) Double-positive HSPCs targeted at *HBB* (GFP) and *CCR5* (mCherry) were sorted at day four post-electroporation and cultured for 15 days while analyzing reporter expression. Error bars represent S.E.M. (*N* = 3 different HSPC donors). (c) Representative gel images showing PCR genotyping of six (out of 57 total) *HBB*-GFP^high^ (gene reporter 1)/*CCR5*-mCherry^high^ (gene reporter 2) methylcellulose-derived clones confirming integration at each locus (d) Representative fluorescence microscopy images of methylcellulose-derived clones with di-genic targeting at *HBB* and *CCR5* show myeloid and erythroid progenitors with both GFP and mCherry expression. (e) HSPCs were targeted at the *HBB* and *AAVS1* loci with a GFP and BFP expression cassette, respectively. Representative FACS plot (left panel) shows analysis seven days after targeting. All four gated populations were sorted and genomic DNA was subject to TIDE analysis for determining INDEL frequencies at the two loci (middle panel), and subject to ddPCR quantification of one of the two possible monocentric translocations between *HBB* and *AAVS1* (right panel) (see also *Figure 3—figure supplement 2*). (f) Representative FACS plots from cells targeted at the *HBB* and *AAVS1* loci with a GFP and BFP expression cassette, respectively. Representative FACS plot shows analysis four days after targeting at which point the four populations were sorted and transplanted intrafemorally into NSG mice that were irradiated 24 hr before transplantation. (g) Bone marrow from the injected femurs from the mice transplanted as described in (f) was analyzed 12 weeks after transplantation. Representative FACS plots are from a mouse from each of the four groups depicted in (f) as well as a mouse transplanted with mock-electroporated cells. The middle row depicts human engraftment gated as positive for the human leukocyte antigen complex (HLA-ABC). The upper and lower rows

*Figure 3 continued on next page*

*Figure 3 continued*

depict FACS plots gated from the human populations and show myeloid (CD33[+]) and lymphoid (CD19[+]) engraftment (upper row) as well as reporter gene expression (lower row) (see also *Figure 3—figure supplement 3* for all transplantation data).

DOI: https://doi.org/10.7554/eLife.27873.009

The following figure supplements are available for figure 3:

**Figure supplement 1.** Cas9 and rAAV6-mediated di-genic homologous recombination (HR) in human CD34[+] HSPCs.

DOI: https://doi.org/10.7554/eLife.27873.010

**Figure supplement 2.** Measuring translocations after *HBB* and *AAVS1* di-genic targeting.

DOI: https://doi.org/10.7554/eLife.27873.011

**Figure supplement 3.** Analysis of mice transplanted with different sorted populations of cells targeted at the *HBB* and *AAVS1* locus.

DOI: https://doi.org/10.7554/eLife.27873.012

human T cells led to translocation frequencies between the two targeted genes of 0.01–1% with monocentric translocations occurring most frequently (*Poirot et al., 2015*), we assessed if our di-genic targeting scheme would enrich for translocations after purification of dual-reporter positive cells. Therefore, we analyzed one of the monocentric translocations between *HBB* and *AAVS1* (*Figure 3—figure supplement 2a*). We targeted *HBB* and *AAVS1* with a GFP and BFP reporter, respectively, and sorted the four different populations (double negative, single positives (each gene), and double positive) seven days after targeting (*Figure 3e*, left panel). INDEL rates at *HBB* and *AAVS1* were comparable among all four sorted populations, with a small enrichment of INDELs in the three populations positive for the reporter (*Figure 3e*, middle panel). Droplet digital PCR (ddPCR) quantification of the translocation showed frequencies ranging from 0.14–0.28%, and importantly, no evidence of enrichment of the translocation was observed in the population sorted for di-genic targeting (*Figure 3e*, right panel and *Figure 3—figure supplement 2c*). Cloning and sequencing of PCR products spanning the translocation showed a wide variety of translocation junctions derived from different DNA end-processing products (*Figure 3—figure supplement 2b*).

To confirm that HSPCs with long-term and multi-lineage engraftment potential were targeted, we again targeted *HBB* and *AAVS1* with a GFP and BFP reporter, respectively, and transplanted the four different sorted populations into immune-compromised NSG mice (*Figure 3f*). 12 weeks after transplantation, human multi-lineage engraftment was evident in the bone marrow of the transplanted mice of all four groups (*Figure 3g* and *Figure 3—figure supplement 3*).

Collectively, these data show that human HSPCs that have undergone di-genic HR are not enriched for translocations, and maintain their multi-lineage colony forming capacity and long-term engraftment potential.

## Multiplexed homologous recombination in HSPCs

We next tested if we could combine the di-genic and biallelic targeting approach to simultaneously target both alleles of *ASXL1* (GFP and mCherry) as well as both alleles of *RUNX1c* (BFP and E2-Crimson) (tetra-allelic) (for schematic see *Figure 4—figure supplement 1a*). Delivery of Cas9 RNPs targeting both genes followed by transduction of four rAAV6 donors gave rise to 1.1% GFP[high]/mCherry[high]/BFP[high]/E2Crimson[high] quadruple-positive cells (*Figure 4a* and *Figure 4—figure supplement 1b–c*). A similar quadruple-positive population was evident when targeting all four combined alleles of *HBB* and *RUNX1c* (*Figure 4—figure supplement 1e–h* and *Supplementary file 1e*). Mixed, myeloid, and erythroid colonies were formed at frequency and ratio comparable to AAV only controls (*Figure 4b*). Genotyping of colonies revealed on-target integration at both alleles at both loci in 78% of clones (73 clones analyzed) (*Figure 4c*). Flow-cytometric analysis of individual colonies confirmed expression of all four reporters (BFP/GFP/mCherry/E2Crimson) at high levels (*Figure 4—figure supplement 1d*). The total number of genetic changes in this enriched population, which could be used for synthetic biology purposes is six: two endogenous genes inactivated (both alleles of each gene) plus the addition of four different transgenes (represented in our experiment by four genes encoding different fluorescent proteins). Thus, this methodology could be used for studying interaction of genes that need both copies disrupted to lose function, such as tumor suppressor genes.

Multi-genic HR in HSPCs would allow for the characterization of functional gene networks during human hematopoiesis (*Bystrykh et al., 2005*). To validate that our methodology could multiplex HR

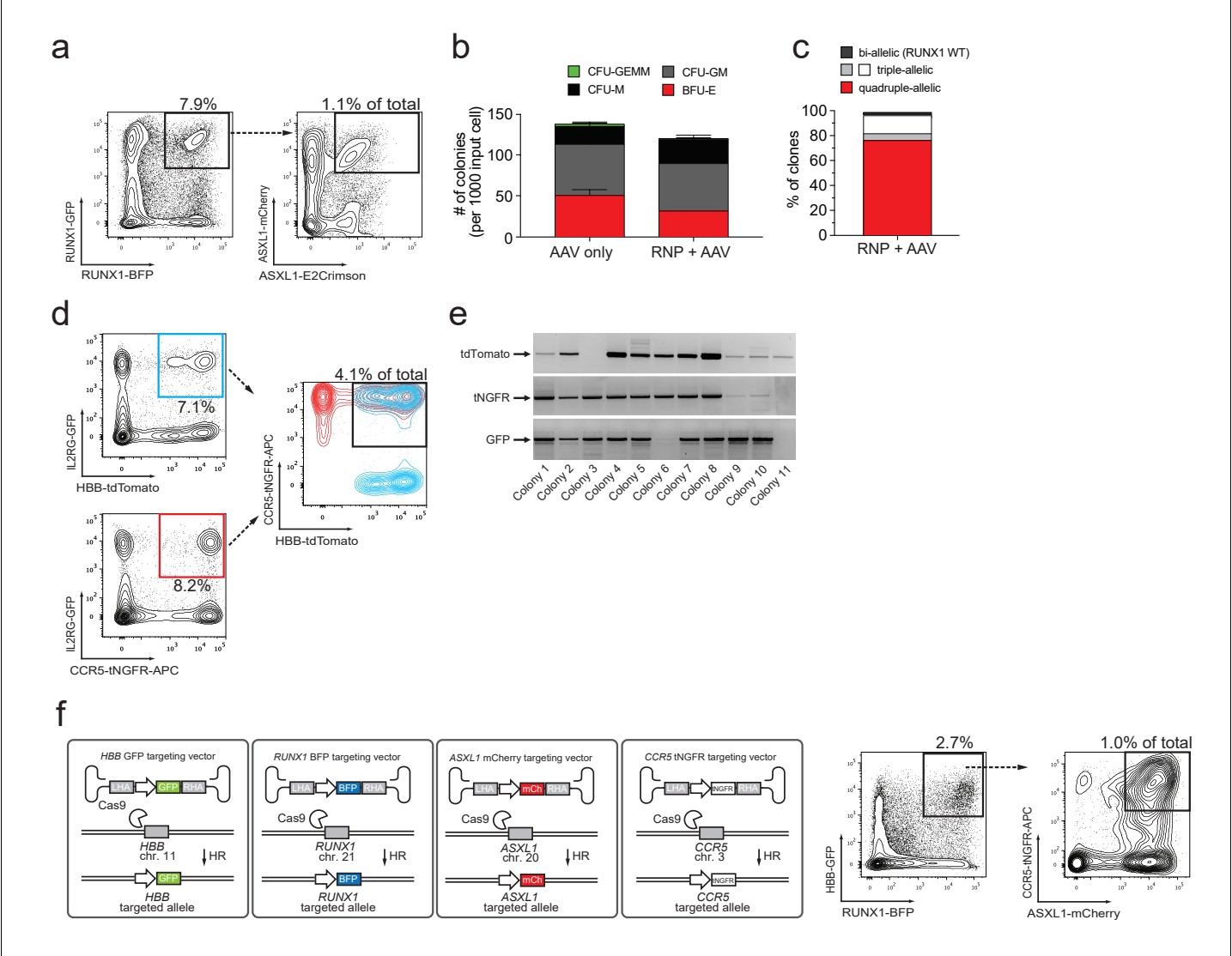

**Figure 4.** Multiplexing homologous recombination in CD34[+] human hematopoietic stem and progenitor cells (HSPCs). (a) HSPCs were electroporated with Cas9 RNP targeting *ASXL1* and *RUNX1* followed by rAAV6 transduction with two donors for *ASXL1* (mCherry and GFP) and two donors for *RUNX1* (E2Crimson and BFP). Tetra-allelically targeted HSPCs were identified as mCherry[high]/GFP[high]/BFP[high]/E2Crimson[high] (*N* = 3 see ***Supplementary file 1e***) (b) Cells modified at both alleles for *RUNX1* and *ASXL1* (as in (a)) were subjected to a methylcellulose assay (triplicates) and scored as BFU-E, CFU-M, CFU-GM or CFU-GEMM based on morphology 14 days after sorting. (c) PCR was performed on colony-derived gDNA to detect targeted integrations at both genes. 73 individual colonies were analyzed. Color coding for colonies with triple-allelic integration are as follows: grey: *RUNX1* biallelic/*ASXL* monoallelic; white: *RUNX1* monoallelic/*ASXL1* biallelic. (d) For tri-genic targeting of HSPCs, cells were electroporated with Cas9 RNP targeting *IL2RG*, *HBB*, and *CCR5* followed by transduction of three rAAV6 donors homologous to each of the three genes (*IL2RG*-GFP, *HBB*-tdTomato, and *CCR5*-tNGFR). Tri-genic-targeted cells were identified as reporter[high] for all three reporters (*N* = 5 see ***Supplementary file 1e***). (e) Methylcellulose clones from the triple-positive cells in (d) were subjected to genotyping PCR and gel images show colonies with targeted integration at all three genes in 9/11 colonies (note that GFP shows a faint band in colony 6). (f) *Left*, Schematic showing strategy for targeting four different genes (*HBB*, *RUNX1*, *ASXL1*, and *CCR5*) simultaneously (tetra-genic). Four different genes are targeted by electroporation of four different Cas9 RNPs followed by transduction with four different rAAV6 donors that each targets a gene with a different reporter. *Right*, Tetra-genic targeting at the above-mentioned four genes was identified as reporter[high] for all four reporters (*N* = 3 see ***Supplementary file 1e***).

DOI: https://doi.org/10.7554/eLife.27873.013

The following figure supplements are available for figure 4:

**Figure supplement 1.** Targeting two genes for biallelic homologous recombination (HR) in primary CD34[+] HSPCs.

DOI: https://doi.org/10.7554/eLife.27873.014

**Figure supplement 2.** Multiplexing homologous recombination at three genes simultaneously in HSPCs.

DOI: https://doi.org/10.7554/eLife.27873.015

*Figure 4 continued on next page*

*Figure 4 continued*

**Figure supplement 3.** Toxicity assessment of multiplexed HR.
DOI: https://doi.org/10.7554/eLife.27873.016
**Figure supplement 4.** Assessment of false-positive frequencies of FACS-based identification of multiplexed HR in HSPCs.
DOI: https://doi.org/10.7554/eLife.27873.017
**Figure supplement 5.** Controlling genotype with cDNA knock-in.
DOI: https://doi.org/10.7554/eLife.27873.018

in HSPCs in more than two genes simultaneously, we electroporated HSPCs with RNPs targeting *HBB*, *CCR5*, and *IL2RG*, and then transduced them with gene-specific rAAV6 donors (*HBB*-tdTomato, *CCR5*-tNGFR, *IL2RG*-GFP) (for schematic see *Figure 4—figure supplement 2a*). At day four post-electroporation, 4.1% of HSPCs were triple-positive (*Figure 4d* and *Figure 4—figure supplement 2b*). 'In-Out PCR' on gDNA from myeloid and erythroid colonies derived from this population showed that 78% (27 clones analyzed) had an integration event at all 3 loci, indicating at least mono-allelic integrations at each targeted locus (*Figure 4e*). Further analyses showed that 85% of these clones with tri-genic integrations were modified on all alleles either by biallelic integration or INDELs on the non-integrated allele that were mostly disruptive (*Supplementary file 1d*). These data confirm that the methodology can efficiently enrich for HSPCs with multiplexed HR. Targeting at another combination of three genes (*RUNX1/HBB/ASXL1*) showed 2.9% triple-positive cells (*Figure 4—figure supplement 2c–e*), and collectively, tri-genic targeting experiments yielded an average of 4.5% triple-positive cells, with the highest frequency of 14% (*N* = 5) (*Supplementary file 1e*). To test if multiplexing HR caused cellular senescence or more cell death than mono or di-genic targeting in HSPCs, we evaluated cell death and apoptosis rates at day three post-targeting and proliferation for up to 10 days post-targeting (corresponding to 7 days post-sorting). We observed similar proliferation rates comparing modified and unmodified cells (data not shown) and only a minor, non-statistically significant decrease in cell viability (p=0.333) when targeting three genes compared to one (*Figure 4—figure supplement 3*). Finally, we targeted HSPCs for tetra-genic HR (*HBB*, *CCR5*, *ASXL1*, *RUNX1*) and found after four days in culture that 1% of cells were reporter$^{high}$ positive for all four reporters (*Figure 4f*). Targeting the same four genes with other combinations of reporter genes gave 0.41% and 0.78% tetra-genic targeting frequencies in the total cell population (*Supplementary file 1e*). Strikingly, 41–71% of HSPCs with tri-genic HR had undergone tetra-genic HR, suggesting that HR events at different genes may not be independent of each other, in contrast to recent findings for multiplexed NHEJ (*Hultquist et al., 2016*). Because rAAV vectors can be captured at DSBs via NHEJ (*Miller et al., 2004*), we performed experiments that aimed to detect the frequency of capture events by including a non-homologous rAAV donor in targeting experiments. We found that 89–98% of reporter$^{high}$ cells were derived from on-target homologous recombination, confirming a relatively low rate of AAV capture (*Figure 4—figure supplement 4*).

## Discussion

*Table 1* summarizes the HR multiplex experiments (seven total genes targeted) and shows that by using Cas9 RNP, rAAV6, and flow cytometry-based sorting, we can reproducibly generate HSPC populations that have undergone HR events at multiple loci. For synthetic biology purposes, the tetra-genic targeting method, for example, can generate an enriched population of cells with eight genetic modifications: the knockout of at least a single allele of four different genes while introducing four different transgenes (in this proof-of-concept we used three fluorescent protein reporter genes and one biologically inert cell surface marker (tNGFR) that has been previously used in human clinical trials to track genetically modified hematopoietic stem cells over the course of decades). Our approach to studying gene function in human HSPCs has several advantages over lentiviral-based approaches because it enables: (1) multigenic targeted integration (at least four genes), (2) enrichment of highly pure edited populations, (3) the ability to trace cells with a specific genotype, (4) enrichment of a population with biallelic targeting of at least two genes, and (5) fluorescent protein-based hematopoietic cell lineage tracing. Our methodology has the potential to advance the biological understanding of gene functions in canonical HSC processes, including self-renewal,

**Table 1.** Overview of targeting experiments in hematopoietic stem and progenitor cells (HSPCs). Overview of all HSPC targeting experiments performed in this study with the number of independent experiments (N) for each experiment type, and the mean targeting efficiency (±SD). See also **Supplementary file 1a, c, and e**.

| Experiment | N | % efficiency ± SD |
| --- | --- | --- |
| Monogenic | 47 | 21.7 ± 13.4 |
| Biallelic | 16 | 5.5 ± 4.2 |
| Di-genic | 17 | 8.1 ± 8.1 |
| Tetra-allelic | 3 | 0.9 ± 0.3 |
| Tri-genic | 6 | 4.5 ± 4.8 |
| Tetra-genic | 3 | 0.7 ± 0.3 |

DOI: https://doi.org/10.7554/eLife.27873.019

differentiation, and engraftment, all of which are critical aspects of fundamental stem cell biology and may augment the efficacy of stem cell based therapeutics.

By knocking in four different transgenes into four different genes, the method generates four gene disruptions and four gene additions. However, the use of multiple sgRNAs also increases the chances for off-target effects and chromosomal translocations. By looking for monocentric translocations between two genes (*HBB and AAVS1*), we observed low levels of translocation events similar to previously published studies (*Poirot et al., 2015*). Such effects are likely sgRNA and target gene-specific and need to be assessed on a case-by-case basis. The observed tetra-genic targeting efficiencies at >0.5% are high enough to be experimentally useful, and though some applications may be restricted by HSPC source and starting cell numbers, our targeting methodology may be combined with recent advances in HSPC expansion protocols (*Fares et al., 2014*; *Cutler et al., 2013*; *de Lima et al., 2012*; *Popat et al., 2015*) or with transplantation into a humanized bone marrow ossicle xenotransplantation model, which supports higher engraftment levels compared to a standard NSG model (*Reinisch et al., 2016*). By using reporters as transgenes, one can both enrich and track the modified cells, and by using a transgene cassette in which a potentially biologically active transgene is linked through a 2A peptide or IRES to a reporter gene, one can enrich and track cells that could have up to four different new potentially bioactive genes expressed. Additionally, we and others have recently demonstrated the feasibility of knocking in a cDNA immediately after the start codon of the gene, thereby maintaining endogenous regulatory control over gene expression (*Dever et al., 2016*; *Hubbard et al., 2016*; *Voit et al., 2014*). This provides a genetic engineering toolbox where different types of alleles (WT, knockout, mutant cDNA forms) are fluorescently tagged and can be enriched or tracked in a population with mixed allele combinations. One potential caveat is the requirement for reporter gene expression and the fact that cells must be cultured for 2–3 days until reporter gene expression is detectable and cells can be sorted. Even though we have not detected any obvious negative impact in this or previous studies (*Dever et al., 2016*; *Bak and Porteus, 2017*), future studies may further investigate and optimize ex vivo culturing conditions, as well as promoter and reporter choice for minimal impact on biology and repopulation potential of edited HSPCs.

Our methodology could be used for the characterization of gene interactions during blood and immune system disease pathogenesis. For example, functional knockouts can be created at one gene (e.g. reporter knock-in into tumor suppressor gene), while introducing disease-causing polymorphisms at another gene (cDNA expression cassette knock-in into proto-oncogene) (see **Figure 4—figure supplement 5** for schematic). For example, Zhao et al., showed that the loss of p53 cooperates with the $Kras^{G12D}$ mutation to promote acute myeloid leukemia (AML) in mouse HSPCs using a retroviral methodology (*Zhao et al., 2010*). Our system could be used to address whether these findings can be translated to human HSPCs by achieving site specific HR that would simultaneously knock out a tumor suppressor (e.g. *TP53*) and drive mutant *KRAS* under endogenous regulatory conditions, instead of using strong constitutive exogenous viral promoters with little control over proviral copy number and heterogeneity of transgene expression. However, in cDNA knock-in experiments, proper expression should always be validated since elements in the adjacent reporter

expression cassette or the lack of UTRs and introns could influence cDNA expression (*Sweeney et al., 2017*). We also show biallelic integration in primary human T cells at *CCR5*, which could be therapeutically applicable for engineering HIV-resistance, where biallelic knockout of *CCR5* could be combined with expression of different HIV restriction factors (*Voit et al., 2013*). Additionally, this approach could be useful to extend recently published studies showing high potency of chimeric antigen receptors (CARs) that were site-specifically integrated into the TRAC gene using CRISPR and AAV6 in primary human T cells (*Eyquem et al., 2017*). Multiplexed gene editing may be used to knock-in different CARs or co-stimulatory ligands into genes that are desirable to knock-out in CAR T cell therapy. We anticipate in the future that multiplexed HR mediated cell engineering will facilitate even more sophisticated uses of synthetic biology-based stem cell therapeutics than the examples we have given. Our methodology should also be widely applicable to other cell types of the hematopoietic system besides HSPCs and T cells, and even to cells of non-hematopoietic origin.

In conclusion, we anticipate that this method will be applicable to studying human hematopoiesis and immune system disease pathogenesis through multiplexed, site-specific genome engineering by HR, which has the potential to lead to new discoveries in human hematopoietic stem cell biology.

## Materials and methods

### AAV vector production

AAV vector plasmids were cloned in the pAAV-MCS plasmid (Agilent Technologies, Santa Clara, CA) containing ITRs from AAV serotype 2 (AAV2). *CCR5*, *IL2RG*, *HBB*, *RUNX1*, *ASXL1*, and *CXCL12* vectors contained an SFFV promoter, a reporter gene such as tNGFR, MaxGFP (or Citrine), BFP, mCherry, tdTomato or E2Crimson and BGH polyA. MaxGFP and Citrine are referred to as GFP throughout. For translocation and NSG transplantation experiments, a UbC promoter (approx. 1200 bp) was used in the *HBB* donor instead of an SFFV promoter. For the T cell experiments, donors carried an EF1α promoter (approx. 1200 bp). The homology arms for *IL2RG, ASXL1,* and *CCR5* were 800 bp, whereas left and right homology arms for *HBB* were 540 bp and 420 bp, respectively. The homology arms for *RUNX1, STAG2,* and AAVS1 were 400 bp. CCR5 donors used in T cell experiments expressed Citrine or mCherry from the PGK promoter and contained 400 bp homology arms. rAAV6 vectors were produced as described with a few modifications (*Khan et al., 2011*). Briefly, 293FT cells (Life Technologies, Carlsbad, CA, USA) were seeded at $13 \times 10^6$ cells per dish in ten 15 cm dishes one day before transfection. Each 15 cm dish was transfected using standard PEI transfection with 6 μg ITR-containing plasmid and 22 μg pDGM6 (gift from David Russell, University of Washington, Seattle, WA, USA), which contains the AAV6 cap genes, AAV2 rep genes, and adenovirus five helper genes. Cells were incubated for 72 hr until rAAV6 was harvested from cells by three freeze-thaw cycles followed by a 45 min incubation with TurboNuclease (Abnova, Heidelberg, Germany) or Benzonase (Thermo Fisher) at 250 U/mL. AAV vectors were purified on an iodixanol density gradient by ultracentrifugation at 48,000 rpm for 2.25 hr at 18°C. AAV vectors were extracted at the 58–40% iodixanol interface and dialyzed three times in PBS with 5% sorbitol in the last dialysis using a 10K MWCO Slide-A-Lyzer G2 Dialysis Cassette (Thermo Fisher Scientific, Santa Clara, CA, USA). Vectors were added pluronic acid to a final concentration of 0.001%, aliquoted, and then stored at −80°C until further use. rAAV6 vectors were titered using quantitative PCR to measure number of vector genomes as described before (*Aurnhammer et al., 2012*).

### CD34[+] hematopoietic stem and progenitor cells

Frozen CD34[+] HSPCs derived from mobilized peripheral blood or cord blood were purchased from AllCells (Alameda, CA, USA) and thawed according to manufacturer's instructions. Fresh CD34[+] HSPCs from cord blood were acquired from donors under informed consent via the Binns Program for Cord Blood Research at Stanford University and used without freezing. Fresh CD34[+] HSPCs from bone marrow were obtained from Stanford BMT Cell-Therapy Facility after informed consent. CD34[+] cells were isolated using a human CD34 MicroBead Kit (Miltenyi Biotec, San Diego, CA, USA). Generally, CB-derived HSPCs perform better in HR experiments. CD34[+] HSPCs were cultured in stem cell retention media consisting of StemSpan SFEM II (Stemcell Technologies, Vancouver, Canada) supplemented with SCF (100 ng/ml), TPO (100 ng/ml), Flt3-Ligand (100 ng/ml), IL-6 (100

ng/ml), UM171 (Stemcell Technologies) (35 nM) and StemRegenin1 (0.75 mM). Mycoplasma contamination testing was not performed. Cells were cultured at 37°C, 5% $CO_2$, and 5% $O_2$.

## T cell isolation and culturing

Primary human CD3[+] T cells were isolated from buffy coats obtained from the Stanford School of Medicine Blood Center using a human T Cell Isolation Kit (Miltenyi) according to manufacturer's instructions. Cells were cultured in X-VIVO 15 (Lonza, Walkersville, MD, USA) containing 5% human serum (Sigma-Aldrich, St. Louis, MO, USA), 100 IU/ml human rIL-2 (Peprotech, Rocky Hill, NJ, USA) and 10 ng/ml human rIL-7 (BD Biosciences, San Jose, CA, USA). T cells were activated directly after isolation with immobilized anti-CD3 antibody (clone: OKT3, Tonbo Biosciences, San Diego, CA, USA) and soluble anti-CD28 antibody (clone: CD28.2, Tonbo Biosciences) for 72 hr. Mycoplasma contamination testing was not performed. T cells were cultured at 37°C, 5% $CO_2$, and ambient oxygen levels.

## Electroporation and transduction of cells

All synthetic sgRNAs were purchased from TriLink BioTechnologies (San Diego, CA, USA). sgRNAs were chemically modified with three terminal nucleotides at both the 5′ and 3′ ends containing 2′ O-Methyl 3′ phosphorothioate and HPLC-purified. The genomic sgRNA target sequences with PAM in bold) were: *HBB*: 5′-CTTGCCCCACAGGGCAGTAA**CGG**-3′, *CCR5*: 5′-GCAGCATAGTGAGCCCA-GAA**GGG**-3′, *IL2RG*: 5′-TGGTAATGATGGCTTCAACA**TGG**-3′, *RUNX1c*: 5′-TACCCACAGTGCTTCA TGAG**AGG**-3′ *ASXL1*: 5′-ACAGATTCTGCAGGTCATAG**AGG**-3′, *STAG2:* 5′-AGTCCCACATGCTA TCCACA**AGG**-3′, AAVS1: 5′-GGGGCCACTAGGGACAGGAT**TGG**-3′. Cas9 protein was purchased from Life Technologies and Integrated DNA Technologies. Cas9 RNP was made by incubating protein with sgRNA at a molar ratio of 1:2.5 at 25°C for 10 min immediately prior to electroporation into CD34[+] HSPCs or T cells. CD34[+] HSPCs were electroporated 1–2 days after thawing or isolation. T cells were electroporated three days following activation. Both CD34[+] HSPCs and T cells were electroporated using the Lonza Nucleofector 2b (program U-014) or 4D (program EO-100) (we have not detected any device-specific differences in electroporation efficiencies) and the Human T Cell Nucleofection Kit (VPA-1002, Lonza) with the following conditions: $5 \times 10^6$ cells/ml, 150–300 μg/ml Cas9 protein complexed with sgRNA at 1:2.5 molar ratio. Following electroporation, cells were incubated for 15 min at 37°C after which they were added rAAV6 donor vectors (generally at an MOI (vector genomes/cell) of 50,000–100,000 for each gene). A mock-electroporated control was included in most experiments where cells were handled the same and was electroporated in the same electroporation buffer, but without Cas9 RNP. For experiments targeting multiple loci, electroporation volume and cell numbers were kept the same as stated above, and 150–300 μg/ml Cas9 RNP and MOIs of 50,000–100,000 were used for each targeted locus, but with no more than a total of 60 ug Cas9 per electroporation and 200,000 vector genomes/cell. All AAV vectors were added simultaneously and directly to the cell culture after which the cells were transferred to the incubator without further manipulation. AAV volume was kept less than 20% of the total culturing volume and medium was either supplemented or replaced with fresh medium after overnight culture.

## Measuring multiplexed targeted integration of fluorescent and tNGFR donors

Reporter[high] expression was measured by flow cytometric analyses after 3–4 days post-electroporation and transduction using gates for multiplexed targeted integration set so that 'AAV only' samples (no nuclease) were less than 1% since previous data (not presented) have shown that after ~14 days in culture the frequency of reporter[+] cells (from persistent episomal expression, random integration, and/or non-nuclease mediated HR) is generally less than 1%. The truncated NGFR receptor (tNGFR) where the cytoplasmic intracellular signaling domain is removed and is signaling incompetent, solely served the purpose of a reporter for targeted CD34[+] HSPCs in indicated experiments (*Bonini et al., 2003*). Targeted integration of a tNGFR expression cassette was measured by flow cytometry of cells stained with APC-conjugated anti-human CD271 (NGFR) antibody (clone: ME20.4, BioLegend, San Diego, CA). For enriching of reporter[high] populations, cells were sorted on a FACS Aria II SORP using DAPI, PI (both Thermo Fisher, 1 μg/ml) or LIVE/DEAD Fixable Cell Stain Kit (Life Technologies) to discriminate live and dead cells according to manufacturer's instructions.

## Scoring, FACS-analysis, and genotyping of methylcellulose colonies

Single reporter[high] cells were either single-cell sorted into 96-well plates (Corning) pre-filled with 100 µl of methylcellulose and water in the outer wells or plated at 500 cells per 6 cm dish with methylcellulose (Methocult, StemCell Technologies). After 14 days, colonies were counted and scored as BFU-E, CFU-M, CFU-GM and CFU-GEMM according to the manual for 'Human Colony-forming Unit (CFU) Assays Using MethoCult' from StemCell Technologies and prior expertise (*Majeti et al., 2007*). For DNA extraction from 96-well plates, PBS was added to wells with colonies, and the contents were mixed and transferred to a U-bottomed 96-well plate. From 6 cm dishes, colonies were picked and transferred to PBS. Cells were pelleted by centrifugation at 300xg for 5 min followed by a wash with PBS. Finally, cells were resuspended in 25 µl QuickExtract DNA Extraction Solution (Epicentre, Madison, WI, USA) and transferred to PCR plates, which were incubated at 65°C for 10 min followed by 100°C for 2 min. For *CCR5*, a 3-primer PCR was set up with a forward primer binding in the left homology arm, a forward primer binding in the insert, and a reverse primer binding in CCR5 outside the right homology arm CCR5_inside_LHA: 5'-GCACAGGGTGGAACAAGATGG-3', CCR5_insert: 5'-AAGGGGGAGGATTGGGAAGAC-3', CCR5_outside_RHA: 5'-TCAAGAATCAGCAATTCTCTGAGGC-3'. For all other genes, gene-specific integration was detected by 'In-Out' PCR using a primer that binds outside the homology arm (HA) and a primer specific for the transgene cassette (insert). *HBB*_outside_LHA: GAAGATATGCTTAGAACCGAGG, *HBB*_insert: ACCGCAGATATCCTGTTTGG *IL2RG*_insert: 5'-GTACCAGCACGCCTTCAAGACC-3', *IL2RG*_outside_RHA: 5'-CAGATATCCAGAGCCTAGCCTCATC-3', *RUNX1*_outside_RHA: 5'- GAAGGGCATTGCTCAGAAAA-3', *RUNX1*_insert: 5'- AAGGGGGAGGATTGGGAAGAC-3', *ASXL1*_outside_RHA: 5'- AAGGGGGAGGATTGGGAAGAC-3', *ASXL1*_insert: 5'- CCTCCCAAGCTGGAACTACA-3'. For detecting IL2RG non-integrated (non_int) alleles the following primers were used: IL2RG_non_int_fw: 5'-TCACACAGCACATATTTGCCACACCCTCTG-3', IL2RG_non_int_rv: 5'-TGCCCACATGATTGTAATGGCCAGTGG-3'. For detecting dual integration of GFP and tdTomato into two *HBB* alleles, a primer in *HBB* outside the right homology arm was used together with either a GFP or tdTomato-specific primer: *HBB*_outside_RHA: 5'-GATCCTGAGACTTCCACACTGATGC-3', GFP: 5'-GTACCAGCACGCCTTCAAGACC-3', tdTomato: 5'-CGGCATGGACGAGCTGTACAAG-3'. Clones with di-genic GFP (*HBB*)/mCherry (*CCR5*) and tri-genic GFP (*IL2RG*)/tdTomato (*HBB*)/tNGFR (*CCR5*) integrations were screened for integrations using the same primers as above. All integrated PCR bands were subjected to Sanger sequencing to confirm perfect HR at the intended locus. For flow-cytometric analysis of colonies generated from cells with quadruple-allelic HR, individual colonies were picked and directly resuspended in FACS buffer containing LIVE/DEAD staining solution (LIVE/DEAD Fixable Near-IR Dead Cell Stain, Thermo). After 30 min incubation (4°C, dark) cells were washed in FACS buffer and subjected to analysis. Dead cells were excluded from analysis based on APC-Cy7 positivity.

## Transplantation of CD34[+] HSPCs into NSG mice

6 to 8 week-old NOD scid gamma (NSG) mice were used (Jackson laboratory, Bar Harbor, ME USA). The experimental protocol was approved by Stanford University's Administrative Panel on Lab Animal Care (IACUC 25065). Four days after electroporation/transduction, different populations of live (DAPI-negative) targeted cells were sorted. Mock-treated cells were also sorted to control for the effect of the sorting procedure. Directly after sorting, cells were transplanted into one femur of sub-lethally irradiated mice (200 rad, 24 hr before transplant). Mice were randomly assigned to each experimental group and analyzed in a blinded fashion.

## Assessment of human engraftment

12 weeks after transplantation, mice were sacrificed, mouse bone marrow (BM) was harvested from the transplanted femur by flushing. Non-specific antibody binding was blocked (10% vol/vol, TruStain FcX, BioLegend) and cells were stained (30 min, 4°C, dark) with monoclonal anti-human HLA-ABC APC-Cy7 (W6/32, BioLegend), anti-mouse CD45.1 PE-Cy7 (A20, eBioScience, San Diego, CA, USA), CD19 APC (HIB19, BD511 Biosciences), CD33 PE (WM53, BD Biosciences), and anti-mouse mTer119 PE-Cy5 (TER-119, BD Biosciences) antibodies, and Propidium Iodide to detect dead cells. Human engraftment was defined as HLA-ABC[+] cells.

## Analysis of HBB-AAVS1 translocations

Genomic DNA was extracted from sorted populations using QuickExtract DNA Extraction Solution. For ddPCR quantification of translocations, ddPCR droplets were generated on a QX200 Droplet Generator (Bio-Rad) according to manufacturer's protocol. Briefly, PCR reactions were set up in a 25 µL total volume per reaction with the ddPCR Supermix for Probes (No dUTP) (Bio-Rad). A HEX reference assay detecting copy number input of the *TERT* gene was used to normalize for genomic DNA input (Bio-Rad: saCP1000100). A custom assay designed to detect the translocations between *HBB* and *AAVS1* consisted of: Forward primer: 5'-TCAGGGCAGAGCCATCTATTGC-3', Reverse primer: 5'-CCAGATAAGGAATCTGCCTAACAGG-3', 5'−6FAM/ZEN/3'-IBFQ-labeled Probe (IDT): 5'-CTTC TGACACAACTGTGTTCACTAGCAACC-3'. The translocation assay was used at a final concentration of 900 nM for each of the primers and a final concentration of 250 nM for the probe. 20 µL of the PCR reaction was used for droplet generation, and 40 µL of the droplets was used in the following PCR conditions: 95° - 10 min, 50 cycles of 94° - 30 s, 57°C – 30 s, and 72° - 2 min, finalize with 98° - 10 min and 4°C until droplet analysis. Droplets were analyzed on a QX200 Droplet Reader (Bio-Rad) detecting FAM and HEX positive droplets. Control samples with non-template control ($H_2O$) or genomic DNA from mock-electroporated samples were included in the entire process. Translocation frequencies were calculated as the translocation copy number per µL divided by the TERT copy number per µL. For sequencing of translocations, PCR products were generated using Phusion polymerase (Fisher Scientific) with the forward and reverse primers listed above for the translocation ddPCR assay. PCR amplicons were gel-purified and cloned into the pMiniT 2.0 plasmid using the NEB PCR Cloning Kit (NEB) according to manufacturer's recommendations. Ligated plasmid reactions were transformed into XL-1 Blue competent cells, plated on ampicillin-containing agar plates, and single colonies were sequenced by MCLAB (South San Francisco, CA, USA) using rolling circle amplification followed by sequencing using the following primer: 5'-ACCTGCCAACCAAAGCGAGAAC-3'.

## Analysis of cell viability and proliferation

Modified cells were FACS-sorted into individual wells of a 96-well U bottom plate and expanded in HSPC retention media (see above) at a density of <100,000 cells per mL. To check viability and proliferation after multiplexed HR, cells from a single well were recovered and a known number of absolute counting beads (CountBright beads, Invitrogen) was added. Cells were stained with Ghost Dye Red 780 (Tonbo Biosciences) for 30 min at 4°C in the dark and analyzed on a FACS-Aria II without further manipulation to reduce potential cells loss. Viable cells were determined as GhostDye Red 780 negative and exact cell counts were assessed through concomitant acquisition of 10,000 beads. Cell counts were calculated based on ratio of beads to cells within the suspension.

## Acknowledgements

ROB was supported through an Individual Postdoctoral grant (DFF–1333-00106B) and a Sapere Aude, Research Talent grant (DFF–1331-00735B) both from the Danish Council for Independent Research, Medical Sciences. DPD was supported through the Stanford Child Health Research Institute (CHRI) Grant and Postdoctoral Award. AR was supported by an Erwin Schroedinger Fellowship from the Austrian Research Council (FWF). MHP gratefully acknowledges the support of the Amon Carter Foundation, the Laurie Kraus Lacob Faculty Scholar Award in Pediatric Translational Research and NIH grant support R01-AI097320, and R01-AI120766. RM gratefully acknowledges the support of the Stanford Ludwig Center for Cancer Stem Cell Research, the Stanford Child Health Research Institute (CHRI), and NIH grant support R01-CA188055. RM is a New York Stem Cell Foundation Robertson Investigator and Leukemia and Lymphoma Society Scholar. We thank David Russell (University of Washington) for the pDGM6 plasmid, the Binn's Program for Cord Blood Research (Stanford University) for cord blood-derived CD34[+] HSPCs, Sruthi Mantri (Stanford University) for isolation of CD34[+] HSPCs from cord blood, and Carmencita Nicolas for help with in vivo experiments. We also thank members of the Porteus and Majeti labs, for helpful input, comments and discussion.

## Additional information

### Competing interests

Ravindra Majeti: Ravindra Majeti has equity and consults for Forty Seven Inc. Matthew H Porteus: Matthew Porteus has equity and consults for CRISPR Therapeutics. The other authors declare that no competing interests exist.

### Funding

| Funder | Grant reference number | Author |
| --- | --- | --- |
| Danish Council for Independent Research | DFF-1333-00106B | Rasmus O Bak |
| Stanford Child Health Research Institute | Postdoctoral Award | Daniel P Dever |
| Austrian Research Council | Erwin Schroedinger Postdoctoral Fellowship | Andreas Reinisch |
| Amon G. Carter Foundation | | Matthew H Porteus |
| Laurie Kraus Lacob Faculty Scholar Award in Pediatric Translational Research | Scholar Award | Matthew H Porteus |
| National Institutes of Health | PN2EY018244 | Matthew H Porteus |
| Stanford Ludwig Center for Cancer Stem Cell Research | | Ravindra Majeti |
| National Institutes of Health | R01-CA188055 | Ravindra Majeti |
| New York Stem Cell Foundation | Robertson Investigator | Ravindra Majeti |
| Danish Council for Independent Research | DFF-1331-00735B | Rasmus O Bak |
| National Institutes of Health | R01- AI097320 | Matthew H Porteus |
| National Institutes of Health | R01-AI120766 | Matthew H Porteus |

The funders had no role in study design, data collection and interpretation, or the decision to submit the work for publication.

### Author contributions

Rasmus O Bak, Daniel P Dever, Andreas Reinisch, Conceptualization, Data curation, Formal analysis, Validation, Investigation, Visualization, Methodology, Writing—original draft, Project administration; David Cruz Hernandez, Data curation, Formal analysis, Investigation; Ravindra Majeti, Matthew H Porteus, Conceptualization, Resources, Formal analysis, Supervision, Funding acquisition, Investigation, Project administration, Writing—review and editing

### Author ORCIDs

Rasmus O Bak (iD) https://orcid.org/0000-0002-7383-0297

Andreas Reinisch (iD) http://orcid.org/0000-0001-9333-7689

Matthew H Porteus (iD) http://orcid.org/0000-0002-3850-4648

### Ethics

Animal experimentation: Animal experiments were performed in strict accordance with the recommendations in the Guide for the Care and Use of Laboratory Animals of the National Institutes of Health. The experimental protocol was approved by Stanford University's Administrative Panel on Lab Animal Care (IACUC 25065).

Decision letter and Author response
Decision letter https://doi.org/10.7554/eLife.27873.024
Author response https://doi.org/10.7554/eLife.27873.025

## Additional files

### Supplementary files

• Supplementary file 1. (a) Overview of Cas9 and rAAV6 mono-genic targeting experiments performed in cord blood (CB), bone marrow (BM), and mobilized peripheral blood (mPB)-derived human CD34$^+$HSPCs. This table summarizes all independent experiments targeting *HBB*, *CCR5*, *IL2RG*, *RUNX1*, *ASXL1*, *STAG2*, and *AAVS1* in HSPCs and the reporter genes used. GFP: green fluorescent protein, tNGFR: truncated Nerve Growth Factor Receptor, BFP: blue fluorescent protein. Efficiencies were averaged across 47 independent experiments, *N* = 47. (b) Overview of genotypes for the non-integrated alleles in mono-genic integration experiments. The three tables show the different INDELs that were identified by Sanger Sequencing of the non-edited allele in mono-genic targeting experiments (*CCR5*, *IL2RG*, and *RUNX1*) used to analyze genotype frequencies shown in *Figure 1—figure supplement 2b and d*. Alleles are grouped into WT (blue), INDELs that preserve the reading frame (red) and INDELs that disrupt the reading frame (green). Note that INDELs that preserve the reading frame can potentially be disruptive depending on the size and location. For example, the 147 bp deletion in *RUNX1* is considered disruptive because of its large size and because it deletes the splice donor site in the intron between exon 2 and 3. For *IL2RG*, one clone was found to have an allele with integration of 230 bp from the donor (at the end of the RHA and 72 bp into the ITR). (c) Overview of di-genic and biallelic targeting experiments in cord blood (CB), bone marrow (BM), and mobilized peripheral blood (mPB)-derived human CD34$^+$HSPCs. This table summarizes the experiments targeting HSPCs for biallelic and di-genic HR and the reporter genes used. GFP: green fluorescent protein, tNGFR: truncated Nerve Growth Factor Receptor, BFP: blue fluorescent protein. Efficiencies were averaged across 16 and 17 independent experiments, respectively, *N* = 16 and *N* = 17. (d) Overview of genotypes for the non-integrated alleles in clones with tri-genic integrations. Each row of the table represents the genotype of a colony established from a tri-genic targeting experiment (*IL2RG*, *HBB*, and *CCR5*). Alleles are grouped into WT (blue), INDELs that preserve the reading frame (red) and INDELs that disrupt the reading frame (green). Note that INDELs that preserve the reading frame can potentially be disruptive depending on the size and location. For *HBB* we identified one clone where *HBD* had been used as repair template and three clones with mono-allelic integration of part of the SFFV promoter indicative of HR events that ended prematurely. (e) Overview of tetra-allelic, tri-genic, and tetra-genic targeting experiments performed in human CD34$^+$HSPCs derived from cord blood (CB), bone marrow (BM), and mobilized peripheral blood (mPB). This table summarizes the independent multiplexing HR experiments performed for tetra-allelic, tri-genic, and tetra-genic targeting and the reporter genes used. GFP: green fluorescent protein, tNGFR: truncated Nerve Growth Factor Receptor, BFP: blue fluorescent protein. Efficiencies were averaged across independent experiments, *N* = 3 (tetra-allelic and tetra-genic) and *N* = 6 (tri-genic).
DOI: https://doi.org/10.7554/eLife.27873.020

• Transparent reporting form
DOI: https://doi.org/10.7554/eLife.27873.021

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
