## [Decision Letter]

Thank you for submitting your article "Multiplexed Genetic Engineering of Hematopoietic Stem and Progenitor Cells using CRISPR and AAV6" for consideration by *eLife*. Your article has been favorably evaluated by Sean Morrison (Senior Editor) and three reviewers, one of whom, Ross L Levine (Reviewer #1), is a member of our Board of Reviewing Editors.

The reviewers have discussed the reviews with one another and the Reviewing Editor has drafted this decision to help you prepare a revised submission.

Although all three reviewers found merit in this work, some specific issues were raised which resonated with all of the reviewers and which need to be addressed. One is the relatively low level of engraftment, which is more striking than with the mock controls, raising concerns as to whether the multiplex gene editing is toxic to cells. This is an important question which needs to be thoughtfully addressed. The other major concern relates to potential off-target effects, and the need to see additional genomic data which addresses this question more comprehensively. Third, the authors should discuss how their method can be applied to biologic studies and what are the novel strengths and limitations to their approach.

*Reviewer #1:*

The authors describe a novel genome editing tool using HR mediated genome editing by combining delivery of Cas9/sgRNA complexes by electrocorporation and delivery of the homologous donor DNA and reporter genes by transduction of cells with a recombinant adeno associated vector.

They use this technology to introduce different modifications such as monoallelic and biallelic modification of a particular gene, but also simultaneous targeting of 4 different genes in human HSPCs – which might have important applications for functional and therapeutic studies in hematological diseases.

The paper does push the envelope further by showing additional technical improvements, which are substantive, above and beyond other reports. However, little biological insight is provided and it would be helpful to show novel biology to make this report of wide interest to the field. Nonetheless, as a Tools and Resources submission, this condition is relaxed a bit as long as the technique provides a rigorous proof of principle.

1) Analysis of clones derived from single cell methylcellulose assays harboring monoallelic integration showed gene specific differences in the modification of non-integrated alleles. What do the authors think was the cause of these gene specific differences? And how could these be influenced?

2) For the biallelic targeted integration in HPSCs, why was the rate of double positivity different in cells that have received RNP+AAV targeting e.g. Runx1 (14%) vs. CCR5 (1.2%)? Is this due to a lower transduction efficiency for the CCR5? Or other mechanisms interfering with HR? And how could this be optimized for CCR5? In Figure 1, the authors should include the data for RUNX1, HBB and CCR5 as well.

3) How did the authors design the sgRNA for introducing biallelic modifications? Were sgRNA on the both strands "reciprocal"? or were the sgRNA on both strands located at different nucleotide sequences?

4) For the transplantation studies, 15 weeks after sorting double positive cells and transplanting them into sublethally irradiated mice, mice were sacrificed in engraftment (CDHLA-ABC+/CD45+) was assessed. After 15 weeks 1.5% of cells showed engraftment. Do the authors have sequential data, how this population changes over time – increases/ decreases?

5) When multigenic HR was performed, was there evidence of NHEJ besides the HR directed repair? And what was its frequency? Was there evidence of gene fusions/ translocations?

6) Cell viability 3 days post targeting 3 genes (RNP+AAV) was <70% compared to 100% in AAV only cells. Since the fraction of triple pos. cells is low (e.g. Runx1/HBB/ASXL1 2.9%), was flow analysis repeated at a later time point to ensure the triple-positive cells were still alive?

7) The authors postulate that multiplex genome editing using their method will enable and facilitate functional studies of gene networks. Nevertheless, simultaneous gene editing of 3 genes showed an efficiency of 2.6% and of 4 genes and efficiency of 1% only. These fractions seem very small. Particularly in the setting of in vivo studies such as a BMT experiment: the authors were able to transplant ~30 000 double positive cells modifying 2 genes simultaneously with a double pos. rate of 10.2.% (di genic approach). If the triple or quadruple positive rate drops to 1%, this would mean to transplant 3 000 cells for a BMT, which seems very low. Please comment on this.

*Reviewer #2:*

This is a nicely designed and executed study using CRISPR tools to introduce reporter genes in human CD34+ HSPCs allowing sorting and/or tracking of cells with inactivation of specific genes.

The authors present an elegant approach that would be useful to model disease relevant inactivating mutations in human HSPCs, and potentially also point mutations (the authors discuss how this might be possible by modifying the CRISPR tools they present but do not actually show data on these strategies).

The data are of high quality, well presented and clearly described. A minor criticism is that the results are presented in a repetitive and a bit monotonous style, but nevertheless clear and with enough detail for others to reproduce.

1) Figure 2 shows low engraftment (in the 0.1-1% range) of the edited cells compared to mock, indicating a potential engraftment impairment (although the mouse numbers are low). Can the authors comment on this?

2) Related to my comment above, the authors could discuss a bit more the potential limitations of their approach. For example, an important point to discuss is the time it would take from the beginning of culture to cell sorting and how this might impact HSPC function.

3) Trypan blue and Annexin staining (Figure 1—figure supplement 3 and Figure 3—figure supplement 3) may not be adequate ways to assess toxicity, as dead cells may be continuously removed from the cultures and not detected with these methods. I think that cell counts depicting the expansion of the edited cells vs. mock or unmodified would be more convincing way to evaluate toxicity of the method.

4) Figure 1—figure supplement 1: the number of untargeted IL2RG alleles without indels is much higher than that of untargeted alleles of CCR5 or other genes. Is this due to lower gRNA activity (reflected by overall lower targeting efficiency in the IL2RG locus?) Can the authors comment on this?

*Reviewer #3:*

In this manuscript, Bak et al. describes a method to perform multiplex homology-directed genome editing of human HSPCs. Using Cas9-sgRNA ribonucleoprotein complex and rAAV as a donor for the homology-directed repair template, they remarkably achieve up to four (potentially more) HDR genome editing events, with little if any modification to a published protocol. Some weaknesses include lack of rigorous assessment of potential negative effects of multiplexed editing on HSPCs function and the genetic consequences. The work also lacks evidence that the functional consequences of multiplex genome editing can be read out even with the potential negative effects the editing protocol may have. Overall, this work could potentially be a highly valuable technique for the hematology field upon clarifying these issues.

1) The chimerism achieved after transplantation of dual-edited cells are substantially lower than mock edited cells with no indication as to whether the cells had multi-lineage reconstitution, raising a concern that the multiplexed editing impaired the function of HSPCs. At least the contribution of dual-edited cells to each hematopoietic lineage should be shown. A concern is that this method uses rAAV6 at a high MOI (50,000-100,000 per gene. With tetra-editing, cells are exposed to 200,000 units of AAV). Is this high level of AAV exposure impairing the reconstitution potential of HSPCs (inflammation response or immune reaction?)? Do cells electroporated with Cas9-sgRNA RNP but not exposed to AAV reconstitute better?

2) The authors should analyze the off-target mutagenesis events. There seems to be a substantial fraction of cells that undergo random integration of rAAV6 reporter, perhaps due to the high MOI of rAAV6 used. For example, Figure 1 shows that 7-8% of cells become reporter^low^, some of which are maintained for long term (Figure 1) indicating that the template DNA integrated into the HSPC genome. I agree that sorting reporter^high^ cells will enrich for cells with HDR editing, but for a technology driven study it will be important to elucidate the nature of these reporter^low^ cells. This will be particularly important for some cell like T-cells (Figure 1) that somehow have lower expression of the reporter genes. It also suggests that a fraction of reporter^high^ cells also have substantial non-targeted integration, which may affect their function. Are these cells enriched for randomly integrated cells, or do they have specific integration into specific loci? Are these cells diluting the episome (so that the frequency of reporter^low^ cells decreases in Figure 1) or these cells dying due to the mutagenic events of random integration?

3) There is little effort to demonstrate the precision of the HDR events other than the "in-out PCR", which only tells that the reporter was integrated into the correct loci but does not tell whether the HDR was precise, or whether off-target cutting occurred, or whether translocation occurred at a level that impedes the usefulness of this protocol. The first point is important given that the authors envision their method could be used for targeted SNP knock-in. Off-targeting editing due to the use of multiple sgRNA is cautioned in the discussion but no data provided to assess whether this is a significant concern or not. The authors should perform sequencing analyses to provide quantitative assessment of precise vs. imprecise HDR editing at the targeted loci, and the extent to which off-target editing/translocation occurs.

4) Although I understand that this is a method paper describing a toolbox to edit the genomes of HSPCs in a multiplexed manner, it is difficult to fully appreciate the potential of this method without a real example of how this could be used. In another word, there is no proof that it is possible to assess the functional consequences of multiplex editing other than analyzing the expression of reporter genes. Any of the concerns raised above (1-3) can impede the potential of multiplex HSPC editing. Can the authors provide evidence that multiplex editing can be used to examine the combinatorial effects of gene editing?

---

## [Author Response]

Although all three reviewers found merit in this work, some specific issues were raised which resonated with all of the reviewers and which need to be addressed. One is the relatively low level of engraftment, which is more striking than with the mock controls, raising concerns as to whether the multiplex gene editing is toxic to cells. This is an important question which needs to be thoughtfully addressed. The other major concern relates to potential off-target effects, and the need to see additional genomic data which addresses this question more comprehensively. Third, the authors should discuss how their method can be applied to biologic studies and what are the novel strengths and limitations to their approach.Reviewer #1: […] The paper does push the envelope further by showing additional technical improvements, which are substantive, above and beyond other reports. However, little biological insight is provided and it would be helpful to show novel biology to make this report of wide interest to the field. Nonetheless, as a Tools and Resources submission, this condition is relaxed a bit as long as the technique provides a rigorous proof of principle.

We thank the reviewer for the supportive words. In the revised manuscript, we now provide an example of how the enrichment technology may be used. We now show that knockout of the cohesin complex member *STAG2* significantly decreases erythroid colony formation in human CD34^+^ HSPCs (new Figure 1). Since the sgRNA used to manipulate *STAG2* has moderate efficiency (compared to other genes presented in our manuscript), the magnitude of the observed erythroid differentiation deficit in knockout cells could only be observed because our presented methodology allowed for the purification of modified cells. This further emphasizes the potential of our presented technology. Knockdown strategies using “INDEL-based” CRISPR/Cas9 approaches without the ability to purify modified cells would most likely miss such an important phenotype.

1) Analysis of clones derived from single cell methylcellulose assays harboring monoallelic integration showed gene specific differences in the modification of non-integrated alleles. What do the authors think was the cause of these gene specific differences? And how could these be influenced?

We believe that the differences observed in the modification of the non-integrated allele in clones with mono-allelic integration could be due to differences in the accessibility of the chromatin, which was recently examined in Chen et al., Nature Comm. 2017, Apr 7;8:14958. This could influence the efficiencies of the individual sgRNAs and dynamics of INDELs vs. HR.

2) For the biallelic targeted integration in HPSCs, why was the rate of double positivity different in cells that have received RNP+AAV targeting e.g. Runx1 (14%) vs. CCR5 (1.2%)? Is this due to a lower transduction efficiency for the CCR5? Or other mechanisms interfering with HR? And how could this be optimized for CCR5? In Figure 1, the authors should include the data for RUNX1, HBB and CCR5 as well.

We have observed that *CCR5* targeting rates, both mono and bi-allelic, are generally lower than at other loci and believe this may be a sgRNA-specific effect that changes the dynamics and spectrum of INDEL creation which may impact HR frequencies. The lack of CCR5 biallelic targeting is also corroborated in Figure 1 where we analyzed allelic frequencies in methylcellulose colonies. Increasing biallelic targeting at CCR5 may be achieved using a different sgRNA, increasing Cas9 concentrations and/or increasing the amount of AAV6 used. However, we believe that optimizing bi-allelic integration into the *CCR5* gene, while interesting from the perspective of generating an HIV-resistant immune system, goes beyond the scope of the work here. As for the data for Figure 1, we have only performed this experiment for *HBB*.

3) How did the authors design the sgRNA for introducing biallelic modifications? Were sgRNA on the both strands "reciprocal"? or were the sgRNA on both strands located at different nucleotide sequences?

For each targeted gene, only one sgRNA is used. This sgRNA targets the same target sites on the two alleles. We have made textual changes to clarify this. Thus, it is important to target sequences that are not polymorphic in the human population when using a single sgRNA and trying to create bi-allelic modifications.

4) For the transplantation studies, 15 weeks after sorting double positive cells and transplanting them into sublethally irradiated mice, mice were sacrificed in engraftment (CDHLA-ABC+/CD45+) was assessed. After 15 weeks 1.5% of cells showed engraftment. Do the authors have sequential data, how this population changes over time – increases/ decreases?

We have now extended the transplantation experiment and show new data in Figure 3, Figure 3, and Figure 3—figure supplement 3. These transplants were performed intra-femorally and at end point upon euthanasia, engraftment was assessed in the transplanted femur. To allow robust assessment of engraftment efficiencies, we did not perform serial aspirates from the transplanted femur. The suggestion to examine engraftment over time is an excellent point, and future studies using significantly larger numbers of mice focused on the engraftment kinetics of genome edited cells will be necessary to investigate this point.

5) When multigenic HR was performed, was there evidence of NHEJ besides the HR directed repair? And what was its frequency? Was there evidence of gene fusions/ translocations?

If the reviewer refers to NHEJ-mediated capture of the AAV donor at on or off target sites, those are events that we have observed. This data is presented in Figure 4—figure supplement 4 where we mix-match nucleases and donors. The frequencies are below 10% of the total targeting frequencies (i.e. 1 out of every 10 targeted cells may have a capture). If the reviewer refers to NHEJ on non-targeted alleles, we have included that data for tri-genic targeting in Supplementary file 4. These data show that 85% of the clones with tri-genic integrations were modified on all alleles either by biallelic integration or INDELs on the non-integrated allele that were mostly disruptive. As suggested by the reviewer, we have now included data quantifying translocations when targeting two genes simultaneously. As expected, we do observe translocations, although at low frequencies (around 0.3%) for a monocentric translocation, which has been reported to be the most frequent translocation type (Poirot et al., Cancer Res. 2015). This new data is presented in Figure 3 and Figure 3—figure supplement 2.

6) Cell viability 3 days post targeting 3 genes (RNP+AAV) was <70% compared to 100% in AAV only cells. Since the fraction of triple pos. cells is low (e.g. Runx1/HBB/ASXL1 2.9%), was flow analysis repeated at a later time point to ensure the triple-positive cells were still alive?

We show survival as well as proliferation of cells targeted with multiplexed HR at time points beyond 4 days post electroporation in methylcellulose and liquid culture experiments. See for example Figure 1, Figure 2, Figure 3, and 3E. Though this data is not comparative, the CFU assay in Figure 4 compares total colony formation of sorted reporter triple-positive cells to cells only transduced with AAV. Finally, long-term survival of modified cells is confirmed in the transplantation data displayed in Figure 3.

To further underscore this point, we have performed viability as well as a cell proliferation assay over a one-week culture period after sorting di-genically targeted cells (day of sort in this specific experiment is defined as Day 0 even though it is actually day 4 after editing), which confirms viability and proliferation capacity of targeted cells equivalent to non-targeted or mock-treated controls. We have added this data from a single cord blood donor to Author response image 1.

7) The authors postulate that multiplex genome editing using their method will enable and facilitate functional studies of gene networks. Nevertheless, simultaneous gene editing of 3 genes showed an efficiency of 2.6% and of 4 genes and efficiency of 1% only. These fractions seem very small. Particularly in the setting of in vivo studies such as a BMT experiment: the authors were able to transplant ~30 000 double positive cells modifying 2 genes simultaneously with a double pos. rate of 10.2.% (di genic approach). If the triple or quadruple positive rate drops to 1%, this would mean to transplant 3 000 cells for a BMT, which seems very low. Please comment on this.

The reviewer brings up an excellent consideration. We have now included a revised transplantation experiment using higher numbers of cells with di-genic targeted integration (new Figure 3). This data includes a mouse with human chimerism of 19.4%. However, we do agree with the reviewer on this important point and have made textual changes to mention ways to overcome this challenge, such as starting with higher cell doses, expansion of modified cells, or transplantation into human bone ossicles, which require very few transplanted cells (Reinisch et al., Nature Med. 2016 Jul;22(39):812-21.). While we believe that we have offered a multiplexing platform to build around, future studies will be needed to address these issues. In addition, while 1% can seem like a small percentage, human leukemias develop from a single cell and yet end up taking over the hematopoietic system. In collaborative work published with the Cleary lab, for example, leukemias developed after transplantation into NSG mice with very low genome editing frequencies (perhaps as low as 0.1%) (Buechele et al., Blood 2015). In those studies, only a single editing event was created (but in a powerful oncogene (MLL)), and the current approach allows the generation of multiple editing events in genes that might have weaker transformative effects than the MLL-AF9 fusion.

Reviewer #2:[…] The data are of high quality, well presented and clearly described. A minor criticism is that the results are presented in a repetitive and a bit monotonous style, but nevertheless clear and with enough detail for others to reproduce.

We appreciate the reviewer’s high praise of our study, as we also believe that it will advance the field for modeling hematological malignancies. We are sorry that we have erred on being monotonous in our presentation, but we felt it was more appropriate to attempt to give a balanced view rather than to overstate our data. In fact, we are tremendously excited about the data and both of our labs have fully embraced the discoveries and approach for studies of leukemogenesis and for developing novel cell based therapeutics.

1) Figure 2 shows low engraftment (in the 0.1-1% range) of the edited cells compared to mock, indicating a potential engraftment impairment (although the mouse numbers are low). Can the authors comment on this?

We have previously described that HSCs are more refractory to HR compared to short-lived hematopoietic progenitors (Dever and Bak et al., Nature 2016). Therefore, higher numbers of modified cells must be transplanted to achieve engraftment frequencies similar to that of unmodified cells. Since we are targeting multiple genes rather than just one, it is not surprising that within a population of cells purified based on HR-mediated marker expression the number of true HSCs is relatively low and that long-term engraftment consequently will be low. To show that higher engraftment is possible by using more cells, we have now updated the transplantation results with data from a new experiment transplanting higher cell numbers (new Figure 3 and Figure 3—figure supplement 3). For one mouse transplanted with enriched di-genically targeted cells we observed 19.4% multilineage chimerism showing that the limiting factor is HSC cell number within the targeted population. While we believe that we have provided enough evidence that multiplexed targeting occurs in HSPCs with long-term and multilineage potential, further studies are necessary to address the caveats of achieving high targeting rates in the most immature HSC compartment with subsequent high engraftment of multiplexed-HSPCs. We have also made text revisions with suggestions on how to overcome low engraftment (higher cell dose, pre-transplant in vitro expansion, and the use of a human bone ossicle transplantation model).

2) Related to my comment above, the authors could discuss a bit more the potential limitations of their approach. For example, an important point to discuss is the time it would take from the beginning of culture to cell sorting and how this might impact HSPC function.

We completely agree with the reviewer and have added this relevant information to the second paragraph of the Discussion.

3) Trypan blue and Annexin staining (Figure 1—figure supplement 3 and Figure 3—figure supplement 3) may not be adequate ways to assess toxicity, as dead cells may be continuously removed from the cultures and not detected with these methods. I think that cell counts depicting the expansion of the edited cells vs. mock or unmodified would be more convincing way to evaluate toxicity of the method.

We agree with the reviewer. Expansion of edited cells is shown in the CFU assay in Figure 4, which compares the total colony formation of sorted reporter triple-positive cells to cells only transduced with AAV.

However, to further support this finding we have performed a cell proliferation assay as suggested by the reviewer using cell counts over a one-week culture period after sorting di-genically targeted cells. This assay confirms viability and proliferation capacity of targeted cells comparable to non-targeted or mock-treated controls. We have added this data from a single cord blood donor to Author response image 2 (day of sort is Day 0).

**Author response image 2. respfig2:** 

4) Figure 1—figure supplement 1: the number of untargeted IL2RG alleles without indels is much higher than that of untargeted alleles of CCR5 or other genes. Is this due to lower gRNA activity (reflected by overall lower targeting efficiency in the IL2RG locus?) Can the authors comment on this?

As the reviewer suggests, this difference in INDEL frequency is likely gRNA and locus specific, and generally must be determined empirically. The *CCR5* guide used in this study creates a very high percentage of INDELs, specifically with a 1bp insertion (Bak and Porteus, Cell Reports, 2017), compared to the *IL2RG* sgRNA.

Reviewer #3:In this manuscript, Bak et al. describes a method to perform multiplex homology-directed genome editing of human HSPCs. Using Cas9-sgRNA ribonucleoprotein complex and rAAV as a donor for the homology-directed repair template, they remarkably achieve up to four (potentially more) HDR genome editing events, with little if any modification to a published protocol. Some weaknesses include lack of rigorous assessment of potential negative effects of multiplexed editing on HSPCs function and the genetic consequences. The work also lacks evidence that the functional consequences of multiplex genome editing can be read out even with the potential negative effects the editing protocol may have. Overall, this work could potentially be a highly valuable technique for the hematology field upon clarifying these issues.

We very much appreciate the reviewer acknowledging that it is remarkable to achieve HR at 4 genes simultaneously and that this work could be a highly valuable technique for the hematology field. We have addressed the reviewer’s concerns below.

1) The chimerism achieved after transplantation of dual-edited cells are substantially lower than mock edited cells with no indication as to whether the cells had multi-lineage reconstitution, raising a concern that the multiplexed editing impaired the function of HSPCs. At least the contribution of dual-edited cells to each hematopoietic lineage should be shown. A concern is that this method uses rAAV6 at a high MOI (50,000-100,000 per gene. With tetra-editing, cells are exposed to 200,000 units of AAV). Is this high level of AAV exposure impairing the reconstitution potential of HSPCs (inflammation response or immune reaction?)? Do cells electroporated with Cas9-sgRNA RNP but not exposed to AAV reconstitute better?

We appreciate the reviewer’s concerns and agree that maintaining HSPC function is of highest importance for this methodology to be useful. We note that in the revised manuscript we have now updated the engraftment data with a new experiment transplanting more cells (new Figure 3 and Figure 3—figure supplement 3). Notably, we can now report a mouse transplanted with di-genically targeted cells that displays 19.4% chimerism and multilineage reconstitution (B cells and myeloid cells). However, we agree that the engraftment capacity of HSPCs edited at multiple loci might be a concern. Lower engraftment of edited and enriched cells compared to unedited cells is something we have observed before (Dever & Bak et al., Nature 2016). This phenomenon is caused by lower targeting rates in HSCs with long-term repopulation capacity (LT-HSCs). Therefore, an edited population contains less LT-HSCs than a non-edited, and it is therefore expected to observe less engraftment with edited populations. With that said, we are currently investigating expansion protocols, injection of a higher numbers of cells, and transplantation into a human bone ossicle xenograft model, but believe those studies are beyond the scope of this manuscript. We do not believe high levels of AAV exposure is influencing HSPC function, as we see equivalent number of progenitor-derived colonies compared to controls (Figure 4).

2) The authors should analyze the off-target mutagenesis events. There seems to be a substantial fraction of cells that undergo random integration of rAAV6 reporter, perhaps due to the high MOI of rAAV6 used. For example, Figure 1 shows that 7-8% of cells become reporter^low^, some of which are maintained for long term (Figure 1) indicating that the template DNA integrated into the HSPC genome. I agree that sorting reporter^high^ cells will enrich for cells with HDR editing, but for a technology driven study it will be important to elucidate the nature of these reporter^low^ cells. This will be particularly important for some cell like T-cells (Figure 1) that somehow have lower expression of the reporter genes. It also suggests that a fraction of reporter^high^ cells also have substantial non-targeted integration, which may affect their function. Are these cells enriched for randomly integrated cells, or do they have specific integration into specific loci? Are these cells diluting the episome (so that the frequency of reporter^low^ cells decreases in Figure 1) or these cells dying due to the mutagenic events of random integration?

The reviewer brings up an excellent point. The reporter^low^ cells contain mostly cells that are not targeted, but also some cells that are targeted (based on consistent reporter expression over time as shown in Figure 1). We have now included data that presents a more in-depth analysis of the reporter^low^ fraction (new Figure 1—figure supplement 1). This data shows that most of the reporter^low^ cells are diluting the episome and becoming negative over time, but some remain positive and even become reporter^high^ because they are in fact targeted. The intensity of the reporter strongly correlates with the propensity to shift to reporter^high^ expression either due to delayed HR or delayed expression. The T cells shown in Figure 2 have low expression because in this particular experiment a weaker promoter (EF1α promoter which is weaker than SFFV) drives the genes encoding fluorescent proteins. We know from our previous studies (Dever & Bak et al., Nature 2016) that the reporter^high^ monogenic targeted HSPCs can engraft long-term in secondary transplants, so it is unlikely that most of the reporter^high^ cells have impaired function and more likely that they have specific integration into the on-target loci (as based on our genotyping data in colonies after 14 days in culture).

While we believe that off-target effects should always be considered when evaluating the results of a genome editing experiment, since we studied 8 different genes in this work to establish a proof of concept, we do not believe that a comprehensive analysis of potential off-target INDELs is within the scope of this manuscript. We did, however, add data on translocation frequencies (Figure 3 and Figure 3—figure supplement 2) to give readers a sense of the relative frequency of such events (rare, accounting for <0.3%). Finally, we are not aware of any study using genome editing in which an off-target INDEL has confounded a biologic finding (in striking contrast to the RNAi methodology) and remain encouraged for the use of this approach. Nonetheless, we agree that unplanned genomic changes (from the Cas9/gRNA, from the AAV, and from simply culturing cells ex vivo) should always be carefully considered when interpreting the phenotypic effect of a genome editing experiment.

3) There is little effort to demonstrate the precision of the HDR events other than the "in-out PCR", which only tells that the reporter was integrated into the correct loci but does not tell whether the HDR was precise, or whether off-target cutting occurred, or whether translocation occurred at a level that impedes the usefulness of this protocol. The first point is important given that the authors envision their method could be used for targeted SNP knock-in. Off-targeting editing due to the use of multiple sgRNA is cautioned in the discussion but no data provided to assess whether this is a significant concern or not. The authors should perform sequencing analyses to provide quantitative assessment of precise vs. imprecise HDR editing at the targeted loci, and the extent to which off-target editing/translocation occurs.

We appreciate the reviewer’s concerns. We and others have previously confirmed that the majority of HR events are perfect/seamless (for example Sather et al., STM, 2015; Dever & Bak et al., Nature, 2016; Bak and Porteus, Cell Reports, 2017). We believe that a full off-target integration analysis is beyond the scope of this study, but as suggested by the reviewer, we have now included data quantifying translocations when targeting two genes simultaneously in HSPCs (new Figure 3). As expected we do observe translocations, although at low frequencies (around 0.3%) for a monocentric translocation, which has been reported to be the most frequent translocation type.

4) Although I understand that this is a method paper describing a toolbox to edit the genomes of HSPCs in a multiplexed manner, it is difficult to fully appreciate the potential of this method without a real example of how this could be used. In another word, there is no proof that it is possible to assess the functional consequences of multiplex editing other than analyzing the expression of reporter genes. Any of the concerns raised above (1-3) can impede the potential of multiplex HSPC editing. Can the authors provide evidence that multiplex editing can be used to examine the combinatorial effects of gene editing?

We developed this method to be able to study genes involved in hematopoiesis that eliminates the need for random integration of lentiviral vectors to reduce gene expression in HSPCs. To exemplify the applicability of studying HSPC biology using targeted integration of a reporter gene combined with the enrichment methodology, we have now included new data (Figure 1) showing that knockout of the cohesion member gene *STAG2* leads to a drastic loss of erythroid colony formation. This highlights that sorting HR-targeted HSPCs can be used to study gene function.